# Investigation on River Thermal Regime under Dam Influence by Integrating Remote Sensing and Water Temperature Model

**Xi Shi, Jian Sun * and Zijun Xiao**

State Key Laboratory of Hydroscience and Engineering, Department of Hydraulic Engineering,
Tsinghua University, Beijing 100084, China; sx1998@outlook.com (X.S.); xzj18@mails.tsinghua.edu.cn (Z.X.)
* Correspondence: jsun@tsinghua.edu.cn; Tel.: +86-010-62797417

**Abstract:** River water temperature (RWT), a primary parameter for hydrological and ecological processes, is influenced by both climate change and anthropogenic intervention. Studies on such influences have been severely restricted due to the scarcity of river temperature data. This paper proposed a three-stage method to obtain long-term daily water temperature for rivers and river-type reservoirs by integrating remote sensing technique and river water temperature modelling. The proposed three-stage method was applied to the Three Gorges Reservoir (TGR) and validated against in situ measured RWTs in the two study sites, Cuntan and Huanglingmiao. The result showed improvements in the method: the quadrate window selection and RWT correction jointly reduce RMSE from 1.8 to 0.9 °C in Cuntan and from 2.1 to 1.2 °C in Huanglingmiao. As a whole, the estimated daily RWT has a consistent RMSE of 1.1–1.9 °C. Meanwhile, by analysing the Landsat-derived daily RWT, we demonstrated that the TGR had a significant impact on the outflow's thermal regime. At the downstream reach of TGR, an apparent increase in RWT in the cold season and interannual thermal regime delay compared to inflow were found with the increasing water level after the dam construction. All the results and analyses indicate that the proposed three-stage method could be applied to obtain long time series of daily RWT and provide a promising approach to qualitatively analyse RWT variation in the poorly gauged catchment for river water quality monitoring and management.

**Keywords:** river water temperature; Landsat; the Three Gorges Dam





## 1. Introduction

Water temperature is a fundamental indicator of water quality in the hydrologic system [1]. It impacts on diverse physical and chemical processes in the water cycle, such as evaporation and oxygen dissolution [2,3]. Variations in water temperature will significantly influence primary production, species migration and eutrophication in an aquatic ecosystem [4–6] and its cooling performance on the urban heat island [7,8]. Evaluating and moderating these influences requires continuously monitoring the long-term water temperature of different water bodies [9]. However, the in situ measured water temperature dataset is under a sparse and coarse spatial distribution compared with other primary factors such as water level and flow velocity [10]. Meanwhile, climate change and anthropogenic intervention unevenly affect the water temperature in both spatial and temporal dimension [11,12], making it complicated to accurately estimate the water temperature either by an environmental or hydrodynamic model in a poorly gauged area. Hence, deriving a sufficient and accurate water temperature data set covering a long period is essential [12].

Thermal infrared (TIR) remote sensing is an alternative and underexplored technique to achieve water temperature data at various scales [13,14]. Among all the TIR platforms, the satellite-borne sensor is eligible to obtain long-term remote sensing data, as it can repeatedly capture images under recurring features [13]. Regarding TIR data sets obtained by satellite, low spatial resolution data have been extensively used to derive sea and lake

surface temperatures; its retrieval method and application was relatively mature [15–17]. In contrast, for rivers and river-type reservoirs, due to the fact of their long and narrow spatial characteristics, higher resolution TIR data were always requisite, and data processing method needed further improvement as the near-bank environment coupled with external disturbance would frequently and severely affect the retrieved RWT's accuracy [13]. Owing to its finer thermal band spatial resolution (60–120 m) and stable temporal availability, Landsat TIR datasets are currently the most widely used remote sensing data sources for retrieving long-term water temperature of rivers and river-type reservoirs [18,19]. Nevertheless, there are still some unfixed or neglected problems during the retrieval process. First, apart from some specific issues, including the stray light of Landsat 8 TIRS and the scan-line corrector (SLC) failure for Landsat 7 ETM+, all Landsat TIR data facing the same situation in that the original TIR imageries (60–100 m) were pre-processed and resampled to 30 m before provision to the users using cubic convolution, a two-dimension interpolation method based on a cubic polynomial kernel equation, and the native resolution TIR data were unavailable [20]. Although this procedure enhances the spatial resolution of the TIR data to the same scale of other visible and near-infrared bands, it introduces uncertainties to more river pixels near the bank, resulting in a boundary effect attached to both sides of the river channel, which will significantly influence the accuracy of the RWT. Some methods including direct removal of one set of data on both sides of the riverbank [21], cubic convolution-based pure-water pixel selection method [20] and thermal sharpening [19], were proposed and utilised in previous papers, but all of them have their limitations. On the other hand, the thermal infrared radiation emitted by water is affected by bridges, passing ships, and flow turbulence in rivers and river-type reservoirs. Directly using the affected Landsat-derived RWT to calibrate with in situ measured RWT will obtain inaccurate errors and further influence the model calibration in the RWT correction step. Most of the current research were relatively random in the selection of the data window, without studying the outcome of different window shapes and scales, and the selection method cannot repeatedly apply to other river channels. Thus, as we focused on obtaining the temporal variation instead of the spatial patterns of RWT, selecting a proper data window that would not be severely affected by random factors and choosing an appropriate method to eliminate the boundary effect were both essential in the entire data processing flow. Furthermore, there is another problem with Landsat-derived RWT that has been indicated but neglected by previous studies, where Landsat 7 ETM+ derived RWT has an obvious systematic error within the high- and low-temperature domains [22]. Although the error may be cause by factors of different aspects and be hard to reveal, it still should be fixed by an existing or empirical regression model in order to derive accurate satellite-derived RWTs.

Generally, after the quality assessment, the amount of available remotely sensed RWTs for each sensor for each year is less than 10. With these sparse data, it is impossible to present the seasonal and monthly variations of RWT or precisely analyse the interannual differences caused by either climate change or human activities. A combination of the river temperature model, remote sensing RWT, additional climatology or hydrological data could provide a way to estimate continuous daily RWT data. In past studies, some river temperature models, including air2stream and CE-QUAL-W2, have been integrated with Landsat-derived river temperature to retrieve daily RWT [20,23]. However, to which degree of anthropogenic or climatic impacts on the water temperature variation these estimated daily RWTs can precisely present is unknown. Among all human and climatic effects, building a dam and constructing a dam reservoir is one of the most critical impacts affecting a river's thermal regime [24–26]. Many studies recognised that the Three Gorges Dam (TGD) would immediately and significantly influence the outflow river temperature patterns in the lower reach under various discharge [26]. Thus, comparing the inflow and outflow thermal regime during each stage of construction and operation over a long time series was an effective way to evaluate and analyse the dam impact on the RWT. In addition, whether a similar attribute shown by the observed data can also be found or

not by integrating the satellite-derived daily RWT could be regarded as an approach to validate the availability of the RWT retrieval method.

Therefore, this research used the case study of the Three Gorges Reservoir on the Yangtze River. The scope of this study included two aspects: (1) developing a three-stage daily RWT retrieval method for rivers and river-type reservoirs, implementing the method using the Landsat 5 TM and Landsat 7 ETM+ TIR data in two study areas, Cuntan and Huanglingmiao, and calibrating the Landsat-derived, air2stream estimated RWT with the in situ measured data in order to evaluate the method accuracy and (2) analysing the impact of the Three Gorges Dam on the thermal regime based on the Landsat-derived and in situ measured RWTs separately and evaluating the practicability of the three-stage method. Overall, this study mainly aimed to develop the remote sensing RWT retrieval method for rivers or river-type reservoirs and assess the method's accuracy and satellite-derived RWT's practicability under the effects of the dam and reservoir.

## 2. Materials and Methods

### 2.1. Study Areas and In Situ Measured Data

The Yangtze River, with a 6300 km mainstream, is one of the longest rivers worldwide. It originates from Qinghai–Tibet Plateau and flows through the majority of provinces in the middle of China and into the East China Sea. The Yangtze Catchment has a river basin of $1.8 \times 10^6$ km$^2$ and is abundant in hydrological and ecological resources. Since there are over 50,000 dams built or under construction within the Yangtze River Catchment [27], the entire drainage basin has significantly been affected by human activities. Among all these dams, the Three Gorges Dam with a water storage capacity of 40 km$^3$ and sitting on the upper reach of the Yangtze River, has already shown a manifest impact on the river's temperature, e.g., algal blooms associated with water temperature have been observed in many tributaries of the TGR in recent years [28]. Two study areas, Huanglingmiao (A) and Cuntan (B), respectively, located downstream and upstream of the TGR, were selected for this research, as shown in Figure 1.

For study area A, in Hubei Province, the Huanglingmiao hydrological station (Huanglingmiao) was chosen as the river temperature measurement point. Huanglingmiao is approximately 12 km downstream of the TGD with a river width of around 600–750 m, while study area B, located near Chongqing City, belongs to the backwater area of TGR. As Figure 1 shows, the Cuntan hydrological station (Cuntan), within study area B, is approximately 7 km downstream from the confluence of Yangtze River and Jialing River. The river width near Cuntan is nearly 850 m. The distance between the two study areas is approximately 580 km.

Two study areas are both in the subtropical zone. The mean atmospheric water vapour content during the summer is around 5 g/cm$^2$ (http://weather.uwyo.edu/upperair/sounding.html). In both areas, the water temperature varies between 10 and 25 °C, and the air temperature hovers between 5 and 35 °C. Their river temperature and air temperature are heavily affected by the monsoon and have evident seasonal variations. The water temperature is lower than the air temperature in the summer while higher in the winter for both areas.

The in situ measured daily river temperature was observed from two hydrological stations Cuntan and Huanglingmiao at 8:00 a.m. (local time) each day ranging from 2004 to 2016. Concerning the daily discharge, since data from the Huanglingmiao station is only available from 2011 to 2013, we used flow data from Yichang station, which is 38 km downstream to the Huanglingmiao station and shares a similar discharge. The mean daily air temperature data were obtained from the China Meteorological Administration (CMA) (freely available at https://data.cma.cn). Data from dozens of stations, including Shapingba and Yichang within study area A and B, were selected and interpolated in order to retrieve the daily air temperature's spatial distribution.

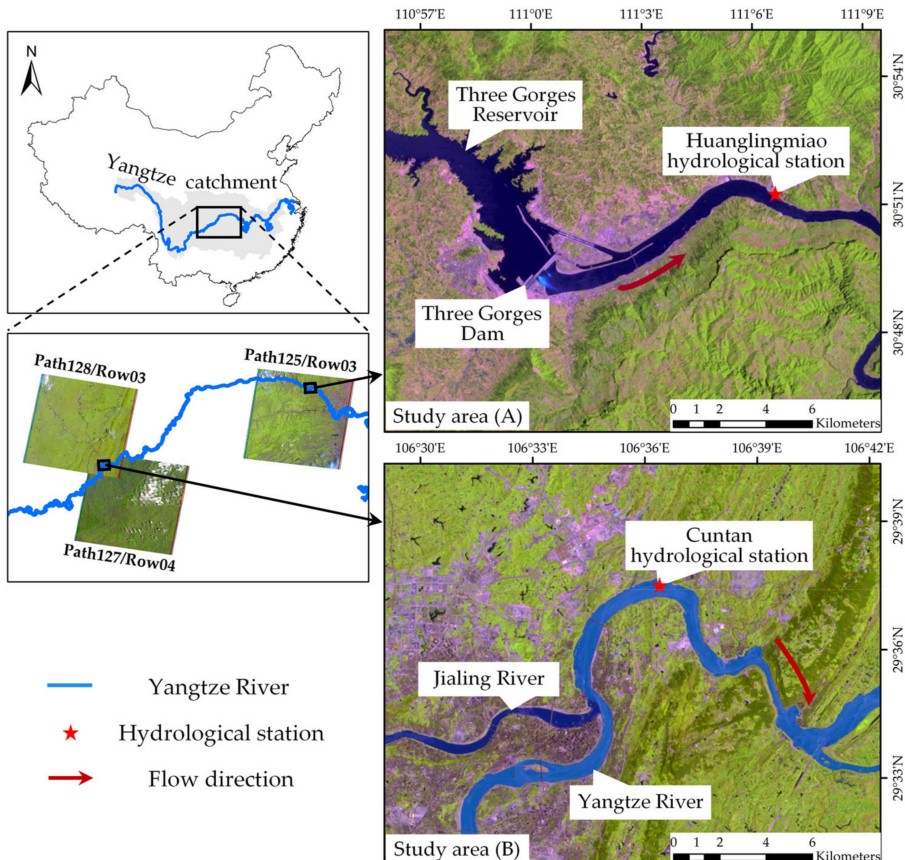

**Figure 1.** Map of the two selected study areas and hydrological stations: (**A**) Huanglingmiao (**B**) Cuntan in the Yangtze Catchment.

### 2.2. Landsat TIR Data Acquisition

Since stray light significantly influences the Landsat 8 TIRS data's accuracy, this study only used Landsat-5 TM and Landsat-7 ETM+ Level 1 datasets including visible, near-infrared and thermal-infrared band images. All these data have a 30 m spatial resolution and 16 days temporal resolution. At the same time, this study also used the LandsatLook Quality Image dataset, which is processed by the US Geological Survey and was mainly used to distinguish and eliminate pixels that were affected by problems such as clouds, cloud shadows, glaciers and geometric correction errors.

Regarding two Landsat datasets mentioned above, this study selected one scene covering study area A, with Worldwide Reference System (WRS) path125/row039 and two scenes overlapping study area B with WRS path127/row040 and path128/row039, as shown in Figure 1. For Landsat 7 ETM+, the SLC-off data were chosen in order to avoid the error caused by the interpolation and gap filling. Geometric and terrain-corrected images from 2004 to 2016 for Landsat 7 ETM+ (the same period as the in situ measured water temperature) and from 2004 to 2010 for Landsat 5 TM (a shorter period because of the termination for Landsat 5) were freely downloaded from the USGS Earth Explorer (https://earthexplorer.usgs.gov/). Generally, these two sensors acquired all the remote sensing images at approximately 11:00 a.m. (local time) each day. All data were under GeoTIFF format with a UTM projection and WGS 1984 datum.

### 2.3. The Three-Stage Landsat TIR Data Processing Method

The Landsat TIR data processing flow was divided into three stages as shown in Figure 2:

1.  Implementing the radiative transfer model (RTM)-atmospheric correction parameter calculator (Atmcorr) model to derive the land surface temperature (LST);
2.  Extracting the RWT from the LST using a Modified Normalised Difference Water Index (MNDWI)-based water mask and correcting the RWT by inviting the in situ measured RWT;
3.  Performing the air2stream model by inputting the Landsat-derived RWT, daily air temperature and discharge to estimate a continuous daily RWT.

The detailed interpretations for each stage are as follows:

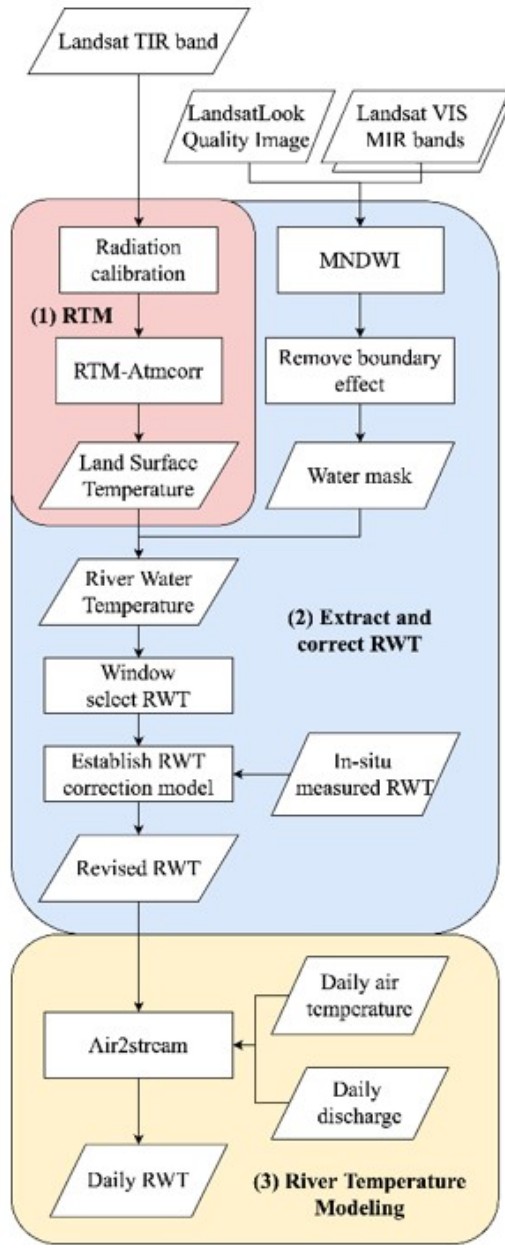

**Figure 2.** The Landsat thermal infrared (TIR) data processing flow chart.

### 2.3.1. RTM

The first step to obtaining the surface temperature is converting the digital number from the original image into the top of atmosphere spectral radiance by carrying out the radiometric calibration, as shown in Equation (1).

$$R_{\text{TOA}} = \text{gain} \times \text{DN} + \text{bias}, \tag{1}$$

where $R_{\text{TOA}}$ is the spectral radiance at the top of the atmosphere (W/sr·m$^2$·nm), DN is the digital number of each raster in the imagery, *gain* and *bias* are the radiometric calibration indicators, respectively; for Landsat 5 TM and Landsat 7 ETM+, high gains were [0.055, 1.18] and [0.037, 3.16], respectively. Different from Landsat 5 TM TIR band, there were two gains for Landsat 7 ETM+ thermal band, high and low gain, respectively, owning a higher radiometric resolution (lower NE$\Delta$T) and a larger radiance detection range, specifically for the extreme hot and cold area [29]. Since the water temperature in both study area is in a normal range, i.e., between 10 and 25 °C, the high gain was used to derive a more accurate RWT.

After the radiometric calibration, another essential step, the atmospheric correction was further implemented in order to retrieve the radiance emitted by the river and land surface. The correction removes the radiance absorbed and emitted by the atmosphere, mainly caused by the water vapour and three primary atmospheric parameters, atmospheric transmittance, upwelling and downwelling spectral radiance, involved in the whole correction process [30,31]. These three atmospheric parameters combined with spectral radiance emitted by the land surface and detected by the sensor at the top of the atmosphere constitute the radiative transfer equation (RTE) [32], which is given as follows.

$$R_{\text{TOA}} = [\varepsilon R_{\text{B}} + (1 - \varepsilon)R_{\text{down}}]\tau + R_{\text{up}}, \tag{2}$$

where $R_{\text{B}}$ is the corrected spectral radiance (W/sr·m$^2$·nm), $R_{\text{TOA}}$ is the spectral radiance detected at the top of the atmosphere calculated in Equation (1), $R_{\text{up}}$ is the upwelling radiance, $R_{\text{down}}$ is the downwelling radiance and $\tau$ is the atmospheric transmissivity; the waterbody emissivity $\varepsilon$ was set as 0.9885 in this paper [33].

Generally, there are two approaches to obtaining the atmospheric parameters: (1) by directly deriving from other additional sources, such as in situ radio soundings and radiative transfer codes [34], and (2) estimating using the empirical equation based on the relationship between atmospheric parameters and water vapour content [35]. Compared to the first method, the second one requires fewer data and effort. Nevertheless, Jiménez-Muñoz indicated that the empirical method is performed imprecisely when the atmosphere is humid, under high water vapour content [36]. Since the water vapour content is extremely high in both study areas during the summer, this study applied the Atmcorr (freely available on https://atmcorr.gsfc.nasa.gov/), which is primarily designed for simulating the atmospheric parameters of Landsat 4–5 TM, Landsat 7 ETM+ and Landsat 8 TIRS thermal bands [37,38]. The Atmcorr database was constructed based on the MODTRAN 4.0 and has a spatial resolution of 1 decimal degree and temporal resolution of 6 h. With all the atmospheric parameters acquired, the corrected spectral radiance was calculated through Equation (2).

For the final step, the corrected spectral radiance was converted to the brightness temperature by Planck's law, as shown in Equation (3).

$$T_B = \frac{K_2}{ln\left(\frac{K_1}{R_B} + 1\right)}, \tag{3}$$

where $T_B$ is the land surface temperature and $K_1$ and $K_2$ are the thermal constants. For Landsat 5 TM and Landsat 7 ETM+, $K_1$ and $K_2$ are 607.76 W/(sr·m$^2$·nm) and 1260.56/K and 666.09 W/(sr·m$^2$·nm) and 1282.71/K, respectively.

### 2.3.2. Generating the Water Mask

The basic approach to generating the water mask in this study was MNDWI [39]. Compared to other methods, such as unsupervised and supervised classification, it was more practical and time-saving to merely employ the Green and MIR bands into Equation (4), sharing a similar accuracy. Meanwhile, different from the initial index NDWI, MNDWI has a better performance when distinguishing water bodies from human artefacts [40], which is more appropriate for the two selected study areas where ships, dams and riverbanks are the most influential objects.

$$\text{MNDWI} = \frac{\text{Green} - \text{MIR}}{\text{Green} + \text{MIR}}, \tag{4}$$

where Green is the digital number of the visible green band (band two), and MIR is the middle-infrared band (band five) of both Landsat 5 TM and Landsat 7 ETM+.

This study set the lower and upper thresholds of the waterbody MNDWI at 0.22 and 0.5, respectively, after a manual MNDWI calculation and comparison. Following the MNDWI waterbody generation, the LandsatLook Quality Image was integrated to eliminate all the pixels with atmospheric and geomorphic problems.

Nevertheless, as mentioned above, all the available Landsat 30 m TIR images were resampled from a coarser original data, 60 m for Landsat 7 ETM+ and 120 m for Landsat 5 TM. Thus, within the water mask obtained by 30 m VIS and MIR bands, there were still two different types of pixels in the river–land boundary with an abnormal temperature value, i.e., Boundary Effect 1 (BE1) and Boundary Effect 2 (BE2) as shown in Figure 3. BE1 represents the 30 m waterbody pixels, which are mixed with the land pixels in the original 60 m spatial resolution. While BE2 are the pixels with an unignored error caused by the cubic convolution resampling (Figure 3b). The temperature of these pixels was more tending to the land nearby and share an excessive error. Therefore, both of them will severely influence the result's accuracy if they were fallen inside the selection window in the next step. In this study, the Boundary Effect 1 and 2 were removed based on the pixel selection method developed by Martí-Cardona, initially designed for the Landsat 8 TIRS thermal bands [20]. In this study, the method was modified to adapt to the Landsat 5 TM and Landsat 7 ETM+ data within the two study areas: (1) A 30–60 and 30–120 m relative position, respectively, for Landsat 7 ETM+ and Landsat 5 TM were assumed. (2) The land-to-water radiance ratio was set at 1.25 instead of 1.2, since the LST in the study area is beyond the RWT. The method is not be fully described in this paper, (refer to Martí-Cardona's paper for more details).

After successfully removing the boundary effects, the ultimate water mask was generated. Then, the LST imagery was subsequently extracted by this water mask, and we obtained the RWT image.

### 2.3.3. Window Select RWT

The spatially continuously distributed RWT needs to be further selected and filtered before it can be calibrated with in situ measured RWT. Currently, it was important to find which selection method to use, as all studies did not use the same approach due to the inconsistency of the different research areas [19,22,41,42]. In order to study the impact of different selection methods on the accuracy of the selected RWT, this paper proposed and compared three different selection methods:

1.  Single-grid selection: Directly recognise the RWT of the raster containing the geographic coordinates of the hydrological station as the final RWT.
2.  Quadrate window selection: First, select a proper size square area, mainly depending on the river width, as the basepoint window centred on the raster containing the geographic coordinates of the hydrological station, as shown in Figure 3c. In this study, we selected a 150 m × 150 m ($L_B = 5$) basepoint window for both study areas. Then, considering each pixel in the basepoint window as the origin, we created a quadrate selection window with side lengths ($L_D$) ranging from 3, 5, and 7 to 43 units,

as shown in the green and blue dashed boxes in Figure 3d, and calculated the average RWT within the window (ignore the raster with Nan value). This creates 25 curves displaying the variation of the mean RWT versus the side length. When the difference between the highest and the lowest value of these 25 RWTs under the same side length is less than 0.1 °C, and as the side length increases, the difference will not be greater than 0.1 °C again, it is deemed that the RWT has converged. The average value of these 25 RWTs is regarded as the final RWT near the hydrological station.

3.  Circular window selection: Change the quadrate window with the side length gradually increasing from 3, 5, and 7 to 43 units in quadrate selection method to a circular window with a diameter ranging from 3, 5, and 7 to 43 units, while maintaining other steps and options the same.

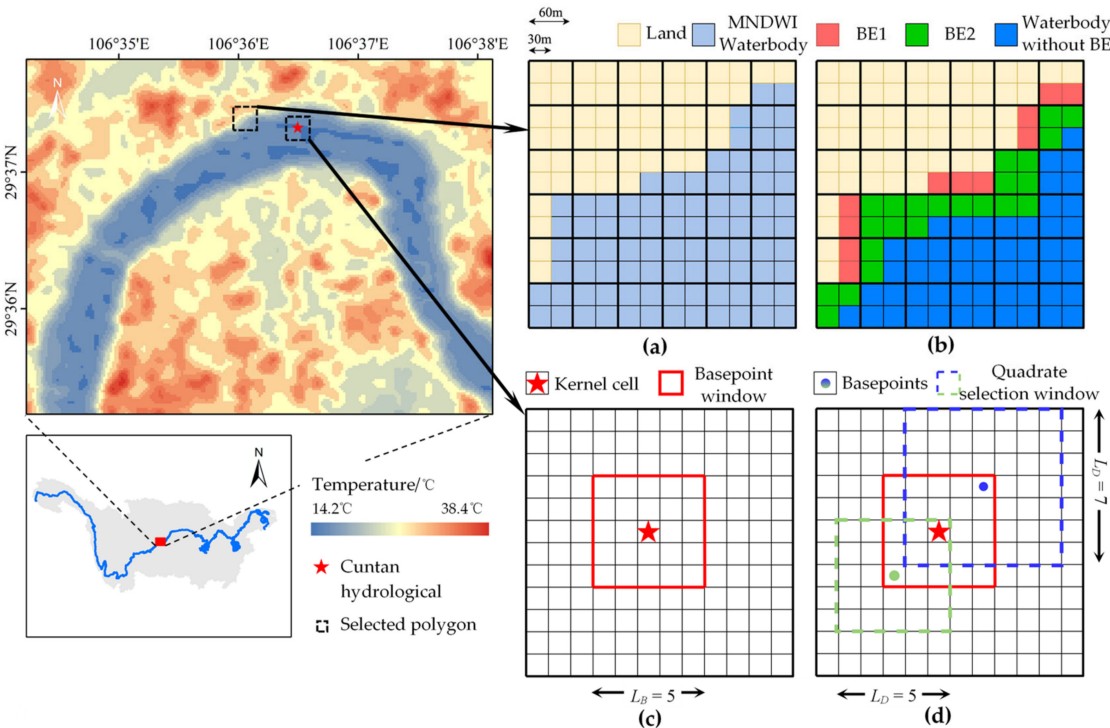

**Figure 3.** Illustrations of two boundary effects: (**a**) Modified Normalised Difference Water Index (MNDWI) waterbody (**b**) Boundary Effect 1 (BE1) and Boundary Effect 2 (BE2) and the quadrate selection method: (**c**) basepoint window and (**d**) quadrate selection window based on Landsat 5 TM-derived river temperature distribution of study area B on 27 October 2012.

### 2.3.4. RWT Correction

As is indicated by previous studies [22] and also shown in Section 3.2, the Landsat-derived RWT has a systematic error potentially caused by compound factors. Thus, in this step, four different methods were executed in this study to correct the RWT, consisting of the modified wave function (WF) model [43], linear, logistic and polynomial regression method. Unlike regression methods, the modified WF model was initially developed to estimate hourly RWT based on the air and water temperature's empirical relationship. In this paper, it was used to remove the RWT error caused by the time gap between the in situ and satellite measurement. The algorithms are shown below:

WF model:

For *sunrise* > 6:00 a.m.

$$T_{corrected} - T_{original} = (T_{max} - T_{min}) \times \cos[\pi \times (\frac{-16}{10 + sunrise} - \frac{H - sunrise - 2}{14 - sunrise})]. \quad (5)$$

For *sunrise* < 6:00 a.m.

$$T_{corrected} - T_{original} = (T_{max} - T_{min}) \times \cos\left[\pi \times \left(\frac{8 - H}{14 - sunrise}\right)\right], \tag{6}$$

where $T_{corrected}$ is the RWT after the correction (°C) and $T_{original}$ is the original RWT (°C), *sunrise* is the sunrise time calculated by sunrise/sunset model employing the date, latitude, longitude and elevation (https://www.mathworks.com/matlabcentral/fileexchange/55509-sunrise-sunset). $H$ is the local time when the satellite captures the imagery. $T_{max}$ and $T_{min}$ are the maximum and minimum RWT simulate from the logistic regression model by inputting the maximum and minimum air temperature.

The regression-based corrections were separately processed for Landsat 5 TM and Landsat 7 ETM+ derived RWT at Cuntan and Huanglingmiao. For all regression methods, around half of the remotely sensed RWT (from 2007 to 2011 for Landsat 7 and from 2006 to 2010 for Landsat 5) and corresponding in situ measured RWT were used to calibrate the parameters in each algorithm. Then, another half dataset was corrected by implementing the calibrated correction equation.

- Linear regression:

$$T_{corrected} = a_1 \times T_{original} + a_2. \tag{7}$$

- Polynomial regression:

$$T_{corrected} = b_1 \times T_{original}^3 + b_2 \times T_{original}^2 + b_3 \times T_{original} + b_4. \tag{8}$$

- Logistic regression:

$$T_{corrected} = c_1 + \frac{C_2 - C_1}{1 + e^{C_3 \times (C_4 - T_{original})}}, \tag{9}$$

where $T_{corrected}$ is the corrected RWT or in situ measured RWT (°C), and $T_{original}$ is the Landsat-derived RWT (°C). $a_n$, $b_n$, and $c_n$ ($n$ = 1, 2, 3, 4) are the regression parameters needed to be calibrated.

### 2.3.5. Air2stream Modelling

For the next step, Air2stream was implemented to derive the continuous daily RWT by inviting the corrected Landsat-derived RWT. Air2stream is a hybrid semiempirical model used for simulating the daily RWT developed by Toffolon and Piccolroaz based on its previous version of air2lake [44,45]. Unlike air2lake, which considers water temperature only related to the air temperature, the new model regards that the heat exchange in the river system is associated with both air temperature and flow discharge. Thus, based on the lumped heat budget equation, an eight-parameter differential equation (Equation (10)) was established to simulate daily RWT, and the eight parameters were calibrated by joining the daily air temperature and discharge combined with the remotely sensed RWT.

$$\frac{dT_r}{dt} = \left(\frac{Q}{\overline{Q}}\right)^{d_4} \times \left\{d_1 + d_2 T_a - d_3 T_r + \frac{Q}{\overline{Q}}\left[d_5 + d_6 \cos\left(2\pi\left(\frac{t}{t_y} - d_7\right)\right) - d_8 T_r\right]\right\}, \tag{10}$$

where $T_a$ is the daily air temperature, $T_r$ is the river water temperature, $t$ is the day of the year (DOY), $t_y$ is the number of intervals during a year, $Q$ is the daily river discharge, $\overline{Q}$ is the average flow during the study period and $d_n$ ($n$ =1, 2, ... 8) are the eight parameters under calibration.

In this study, the TGR was under three separate impoundment and operation periods from 2004 to 2016 with different water levels at the head of the reservoir and impoundage: Stage I from 2004/01 to 2006/05 with a maximum water level of 140 m and a maximum impoundage of $1.63 \times 10^{11}$ m$^3$, Stage II from 2006/09 to 2008/05 with a maximum water level of 155 m and Stage III from 2008/09 to 2016/12 with a maximum water level of approximately 170 m and a maximum impoundage around $3.93 \times 10^{11}$ m$^3$. As impoundage

increased, the thermal regime of inflow and outflow in the different stages was distinct. Hence, the air2stream was separately calibrated and validated under four different conditions: (S1) the inflow (Cuntan or study area B) of TGR during Stage I, (S2) the inflow of TGR during Stage III, (S3) the outflow (Huanglingmiao or study area A) of TGR during Stage I and (S4) the outflow of TGR during Stage III. For calibration of the air2stream model, we set Crank–Nicolson as the method to solve the equation and minimise the error by particle swarm optimisation.

## 3. Results

After the quality assessment, there were 28 Landsat 5 and 68 Landsat 7 valid images at Cuntan, while there were 37 Landsat 5 and 77 Landsat 7 valid images for Huanglingmiao ranging from 2004 to 2016. On average, 5–11 images were available to retrieve the RWT in each year (theoretically, 44–66 images per year). Overall, the validity of the remote sensing data was relatively low, and the detailed results for each stage of this research will be demonstrated in the following sections.

### 3.1. Comparison of RWT with and without the Boundary Effect

In order to present the impact degree of the boundary effect, two RWT line charts with and without boundary effects selected by the quadrate window of Cuntan 20040421 and Huanglingmiao 20060515 are displayed in Figure 4. As is shown in Figure 4a, with the influence of the boundary effect, when the side length $L_D$ of the quadrate window increased to 7 units, the average water temperature in the window began to increase due to the incorporation of high-temperature pixels near the riverbank. When $L_D$ expanded to 27, the uplifting trend slowed down, and the mean RWT in each window began to gradually converge. However, it seems like the RWT did not fully converge, as even the side length increased to 43 units because an increasing number of abnormal pixels (BE1, BE2) were continuously merged in, and the average value of 25 RWTs ($RWT_{BE}$) under 43 units was 20.63 °C, i.e., 0.83 °C higher than the in situ measured RWT ($RWT_M$). In contrast, after removing the boundary effect, when the window side length $L_D$ increased to 23, the average water temperature in the window was already under convergence. The difference between the remotely sensed RWT ($RWT_{NBE}$) was 19.77 °C, and the in situ measured river temperature was only 0.03 °C, which is far less than 0.83 °C. The same condition occurred in Huanglingmiao on 2006/05/15 where the $RWT_{BE}$ owned a larger error ($\Delta = 0.71$ °C) than $RWT_{NBE}$ ($\Delta = 0.42$ °C) compared to the $RWT_M$ as shown in Figure 4b.

Then, 68 valid Landsat-derived RWT images of Cuntan and 77 images of Huanglingmiao before and after removing the boundary effects were utilised to calibrate with in situ measured RWTs. The root-mean-square error (RMSE) decreased from 1.93 to 1.61 °C, accompanied by an increase in the $R^2$ from 0.88 to 0.9 with the removal of the boundary effect in Cuntan, and for Huanglingmiao, the RMSE improved from 2.12 to 1.96 °C. However, as we removed the boundary effect, there was at least 1 to 1.5 pixels under the original thermal band spatial resolution removed at both sides of the river. Therefore, this method requires that the river width to be larger than 240 m for Landsat 7 ETM+ (480 m for Landsat 5 TM) to ensure the RWT's accuracy.

Meanwhile, the error achieved using three different selection methods was also analysed by applying 42 sets of Cuntan Landsat 7 data and 55 sets of Huanglingmiao Landsat 7 data (due to the SLC failure, some images had a Nan value using single-grid selection). Quadrate and circular selection methods, with RMSEs of 1.692 and 1.694 °C in Cuntan and 1.983 and 1.982 °C in Huanglingmiao, shared a better performance, i.e., RMSE = 1.783 °C for Cuntan and 2.129 °C for Huanglingmiao, than the single grid method. This indicates that the random factors within the river would badly affect the data's accuracy. Thus, the quadrate selection method was executed for retrieving Landsat-derived RWT in this study. However, this selection method also has limitations. It can only be used in the river section where water temperatures do not vary sharply along the river, in other words, the river should be well mixed in the study area.

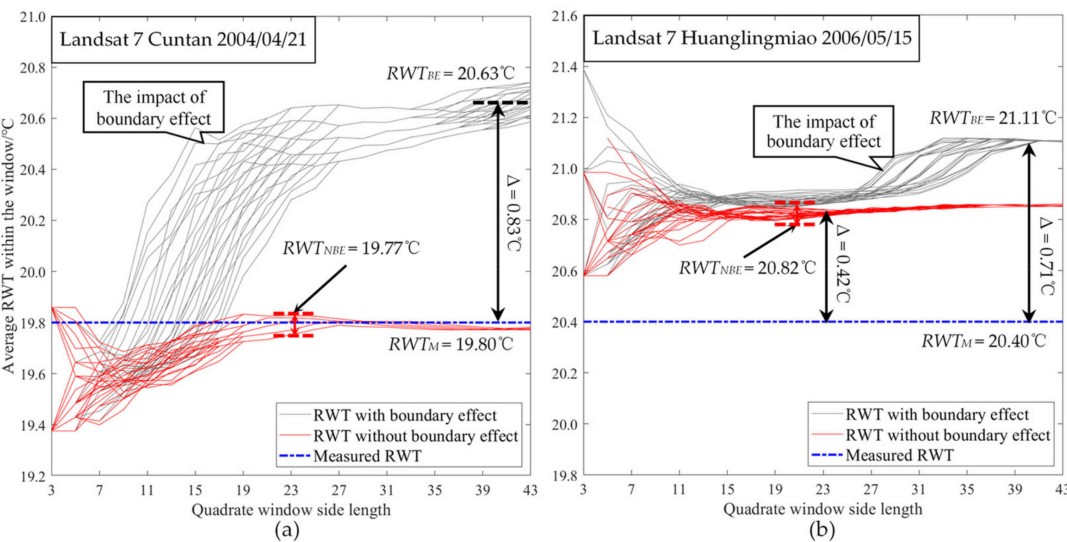

**Figure 4.** Impact of the boundary effect on Landsat 7 ETM+-derived river water temperature (RWT) for (**a**) Cuntan 2004/04/21 and (**b**) Huanglingmiao 2006/05/15.

### 3.2. Corrected RWT

After removing the boundary effect and filtering by the quadrate selection method, the RWTs of Cuntan and Huanglingmiao derived from the Landsat 5 TM and Landsat 7 ETM+ were calibrated with the field surveyed RWTs. The results show that the RMSE of RWT at Cuntan retrieved by Landsat 7 ETM+ was 1.61 °C and that of Huanglingmiao was 1.81 °C; the RMSE of the RWTs of Cuntan and Huanglingmiao obtained by Landsat 5 TM were 0.91 and 1.03 °C, respectively. The accuracy of the remotely sensed RWTs varied tremendously between the different satellites and sensors but shared same accuracy within different study areas. These phenomena indicate that there could exist a systematic error within the Landsat 7 ETM+-derived RWTs. Thus, we scatter plotted the Landsat-derived RWTs with the in situ measured RWT as shown in Figure 5. A considerable part of the grey circles, which denote the original RWT derived from Landsat 7 ETM+, fall outside of the ±1 °C domain of in situ measured RWTs. These kinds of data are mainly located in two regions: high- and low-temperature regions. In the high-temperature domain (in situ measured RWT > 20 °C), the RWT obtained from the satellite data is generally higher than the in situ measured RWT, while in the low-temperature domain (in situ measured RWT < 15 °C), the condition is reversed, resulting in the scatter plot tilting towards the *y*-axis.

Thus, the RWTs derived from both sensors in two study areas were corrected based on three regression methods and the WF model. All the results are shown in Table 1. For Landsat 7, despite the modified WF model, which makes the data more imprecise, all the regression methods based on the empirical relationship between half of the remotely sensed and in situ measured RWT dataset effectively reduced the error of another half. The negative correction of the modified WF model may be caused by two reasons: (1) the modified WF model was not suitable for all catchments, and the model should also be calibrated before the usage and (2) the error of satellite-derived RWT was orientated from multiple sources, which cannot simply be regarded as the heat change during the time interval of in situ and satellite measurement. Among three regression approaches, logistic regression performed the best, decreasing the RMSE of the Landsat 7-derived RWT from 1.621 to 0.856 °C at Cuntan and from 1.989 to 1.213 °C at Huanglingmiao, as shown in Figure 5. Clearly revealed in Figure 5a,b, the original RWT (i.e., the grey circle), especially in the high- and low-temperature regions, was sharply corrected into red points, and most of the logistic-corrected RWT shifted into the ±1 °C domain of the in situ measured RWT. However, for Landsat 5, all the regression methods executed a negative correction on RWT, as shown in Table 1. This indicates that Landsat 5 did not share the same systematic error

with Landsat 7. Therefore, the original Landsat 5-derived RWT were used in the following air2stream modelling.

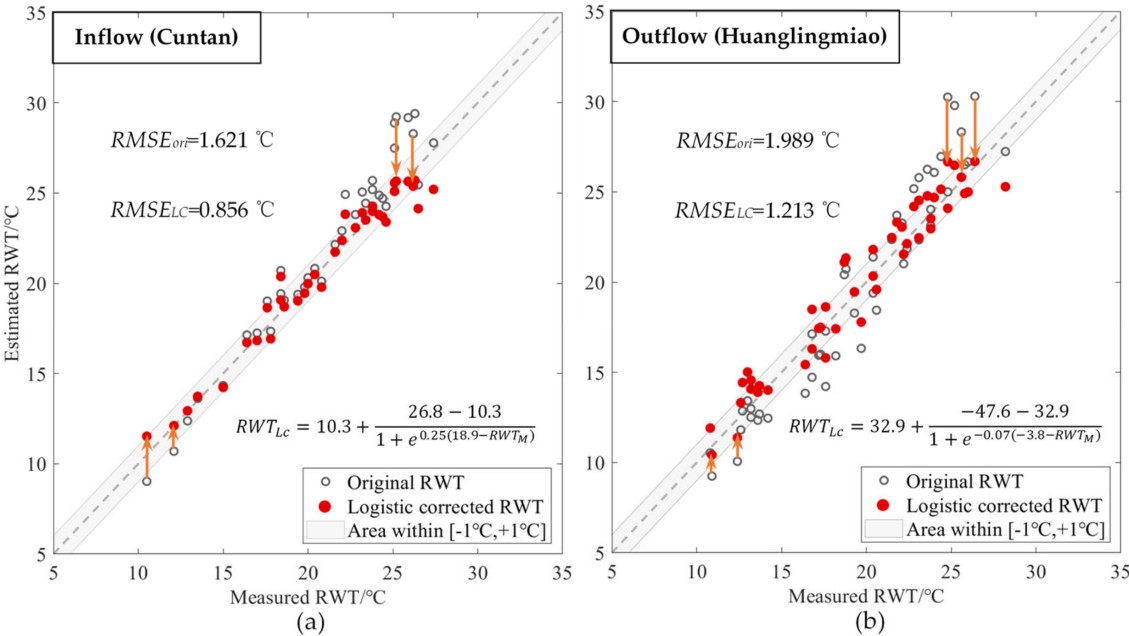

**Figure 5.** Comparison of the original and logistic-corrected RWT: (**a**) inflow (Cuntan) and (**b**) outflow (Huanglingmiao) derived from Landsat 7 ETM+ ranging from 2004 to 2006 and 2012 to 2016.

**Table 1.** The root-mean-square error (RMSE) for original and corrected river water temperature (RWT) within two study areas for Landsat 5 TM and Landsat 7 ETM+.

| Study Area | Sensor | Original RWT RMSE/°C | Corrected RWT RMSE/°C | | | |
|---|---|---|---|---|---|---|
| | | | Linear Regression | Logistic Regression | Polynomial Regression | WF Model |
| Inflow | Landsat 5 TM | 0.564 | 0.874 | 0.862 | 0.682 | 1.257 |
| (Cuntan) | Landsat 7 ETM+ | 1.621 | 0.938 | 0.856 | 0.929 | 2.280 |
| Outflow | Landsat 5 TM | 1.052 | 1.095 | 1.158 | 1.067 | 1.932 |
| (Huanglingmiao) | Landsat 7 ETM+ | 1.989 | 1.400 | 1.213 | 1.215 | 2.937 |

Since the systematic error was only found in Landsat 7-derived RWT, after analysing the data processing flow, we recognised that there may be two reasons causing the problem: (1) a specific issue occurring on Landsat 7 such as the scan-line corrector failure and (2) employing inaccurate calibration or correction parameters in the radiometric calibration or atmospheric correction step. This may be an area for the future research.

### 3.3. Air2stream Estimated Daily RWT

All of the logistic-corrected RWTs from four different conditions, mentioned in Section 2.3.5, were separately imported into the air2stream model along with the corresponding daily air temperature and discharge to derive a continuous daily RWT. The results are shown in Figure 6. Most of the logistic-corrected RWT were close to the simulated daily RWT curve, not only for calibration period but also for the validation period, except for the outflow (Huanglingmiao station) in the latter half of 2006, mainly because, in this period, the construction of TGD was finished and it was planned to impound water in the TGR from a level of 140–170 m for the first time. Thus, all data in 2006 were excluded in the following section.

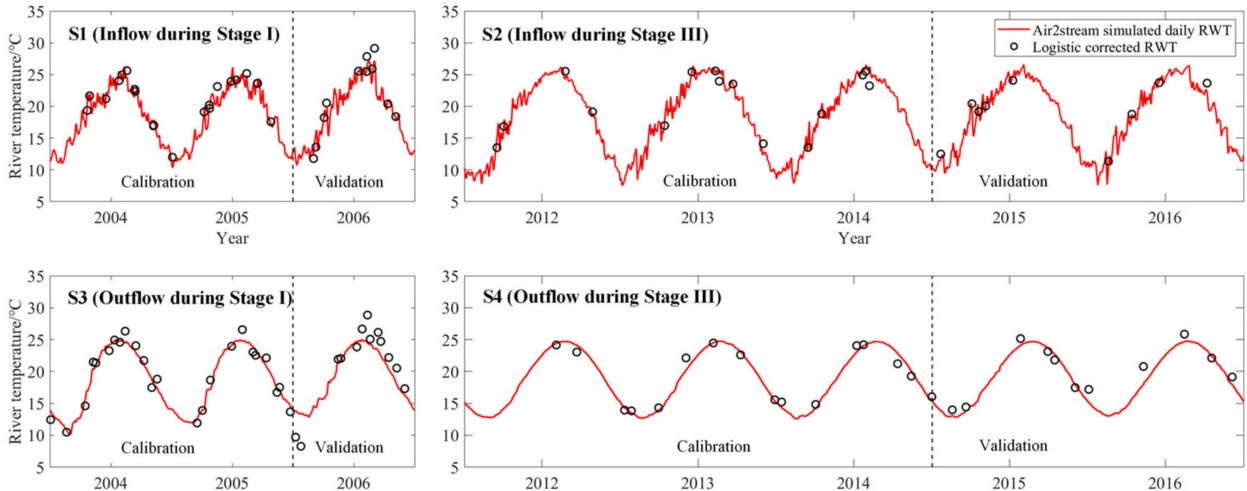

**Figure 6.** Relationship between logistic-corrected RWT and air2stream estimated daily RWT in different conditions.

The RMSE of the estimated daily RWT under each situation are presented in Table 2. Except for the outflow in 2006, the RMSE during Stage I of 1.087–1.485 °C was entirely smaller compared to Stage III with an RMSE of 1.441–1.778 °C, mainly due to the difference of in the amount of remotely sensed data between the two stages. The quantity of available data in Stage I was approximately twice as large than in Stage III, for Landsat 5 terminated at the beginning of Stage III. Meanwhile, there was no manifest difference between the two study areas. Thus, the amount of employed remotely sensed data will severely influence the estimated daily RWT's accuracy. Overall, the air2stream model performed stably and consistently during these four periods and the Landsat-derived air2stream estimated RWT had an RMSE of 1.087–1.778 °C.

**Table 2.** The RMSE of air2stream simulated daily RWT and the available quantity of remotely sensed data per year in the calibration and validation periods during Stage I and III.

| | RMSE/°C (Remotely Sensed Data Amount Per Year) | | | |
| --- | --- | --- | --- | --- |
| | **Stage I** | | **Stage III** | |
| **Study Area** | **Validation (2004–2005)** | **Calibration (2006)** | **Validation (2012–2014)** | **Calibration (2015–2016)** |
| Inflow (Cuntan) | 1.087 (10.5) | 1.485 (11) | 1.574 (5) | 1.778 (5) |
| Outflow (Huanglingmiao) | 1.291 (11) | 2.576 (11) | 1.441 (5.3) | 1.580 (5.5) |

## 4. Discussion

In this section, we discuss the outcome of the Landsat-derived RWT and evaluate the retrieval method from two different aspects: (1) the variation in the outflow RWTs in different impoundment periods and (2) the interrelationship between inflow and outflow during different periods.

### 4.1. Outflow RWT Variations for Different Stages

The operation scheme of the TGD in separate stages will directly influence the outflow discharge and RWT. Thus, the variation of the outflow RWT between Stage I and Stage III were firstly assessed by generating the cumulative probability curve and probability density histogram based on the estimated and in situ measured RWT data separately. As is shown in Figure 7b, compared to Stage I, the in situ measured outflow of RWT in Stage III showed a noticeable phase shift towards the higher end when the RWT was lower than

20 °C indicating that in the cold season, the RWT increased as the maximum water level of TGR rose. A distinct phenomenon can be observed in the warm season (RWT > 20 °C) where the curves in two stages mix. For the Landsat-derived daily RWT, although a similar RWT gap between two different stages can also be found in the low-temperature region and two curves gathering in the high-temperature region, there was a distinct RWT domain and density distribution between estimated and in situ measured RWT. Compared to the in situ measured data, the estimated RWT had a narrow domain in both stages with more data gathering near the lower and upper thresholds, resulting in a steeper curve at both ends as shown in Figure 7a and density peaks appearing in Figure 7c. The primary thought is that the overcorrection of the remotely sensed data caused this phenomenon. Thus, we simulated another set of daily RWTs based on the original Landsat-derived RWT. However, the trend for both curves remained unchanged, and only a 2 °C increment could be found for the lower and upper thresholds of the curve during Stage I. Meanwhile, as is shown in Figure 7a, the curve missed more characteristics in Stage III than Stage I, as the features of Stage III in the high- and low-temperature regions were totally lost, probably because, for each year in Stage I, there were more remotely sensed data added into the air2stream model. All the results indicate that as more Landsat-derived RWT is input into the air2stream model (especially, for the extreme temperatures), the more the characteristics will be represented.

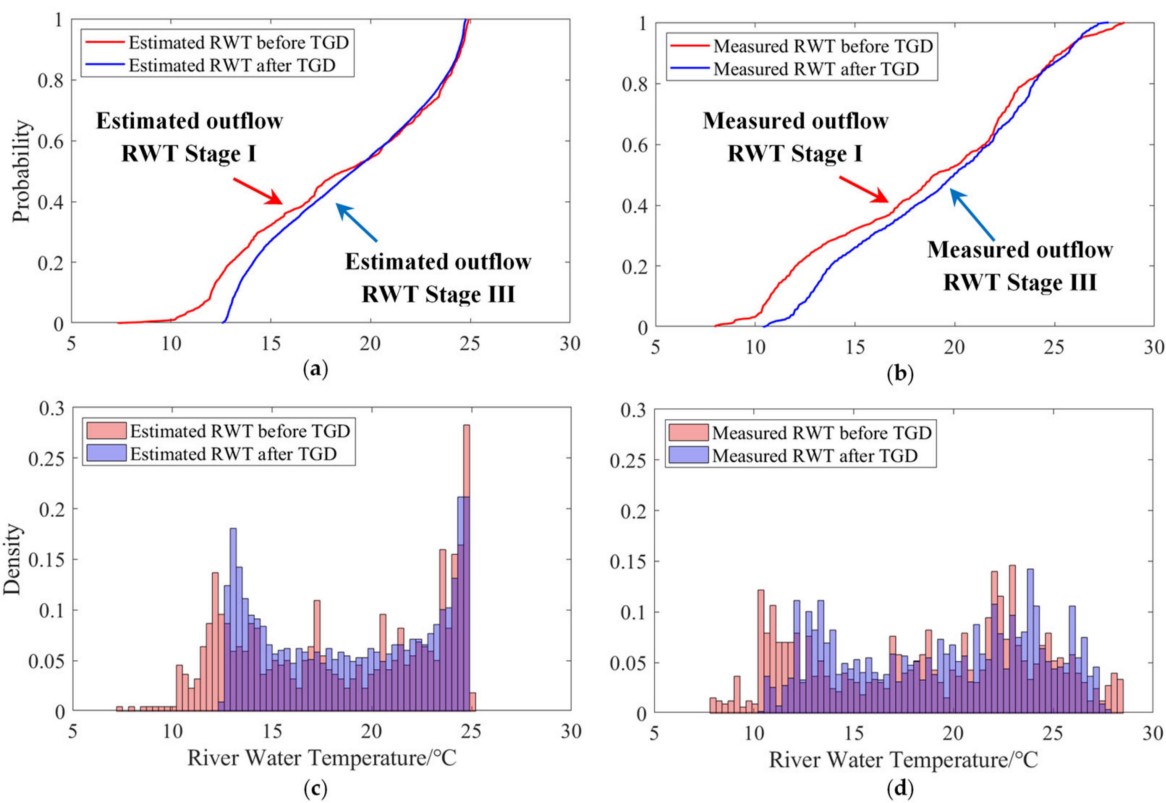

**Figure 7.** Cumulative probability curves of: (**a**) air2stream estimated RWTs during Stage I and Stage III, (**b**) in situ measured RWTs in Stage I and Stage III, (**c**) density distribution of air2stream estimated RWTs in Stage I and Stage III and (**d**) in situ measured RWT in Stage I and Stage III.

### 4.2. Inflow and Outflow RWT Variations

Generally, construction of the reservoir will influence the interrelationship between inflow and outflow RWTs, where the thermal regime of the downstream area will gradually be delayed compared to the upstream due to the dam retention impact. Thus, we generated the inflow and outflow RWT interrelationship curve of the TGR through estimated and

in situ measured RWTs during Stage I (2004–2005) and Stage III (2012–2016) as shown in Figure 8a,b, respectively. In the Stage III, it is evident that the in situ measured RWT of the outflow (Huanglingmiao) increased more slowly than the inflow (Cuntan) from the cold season to warm season while also decreased slowly when returning back to the cold season, as shown in Figure 8b, resulting in a hollow area in the middle of the curve. This delay phenomenon was inconspicuous during Stage I where the hollow area is small and the curve still assembled near catacorner, as shown in Figure 8a. The discrepancy in two figures indicated that the TGD further affected the thermal regime as the increment of impoundage. On the other hand, the same appearance can be found under the estimated RWTs in both figures. However, it seems like the estimated RWT curve had a clockwise rotation compared to the observed RWT curve, mainly caused by inaccurate estimation of the high and low temperatures of the outflow, which was also found in Section 4.1. Therefore, in the current stage, using this data, we can only qualitatively analyse and present the RWT characteristic under the impact of the dam. Meanwhile, all the results of the analyses demonstrated that the three-stage method was precise enough to obtain RWT and part of its features in the river and river-type reservoir and could be applied further in other catchments. However, all the results indicated that extending the remotely sensed data, either by integrating more data from the Landsat mission or other platforms, is an urgent affair for future study if we want to quantitatively analyse the dam's impact, e.g., lag time between different impoundment periods.

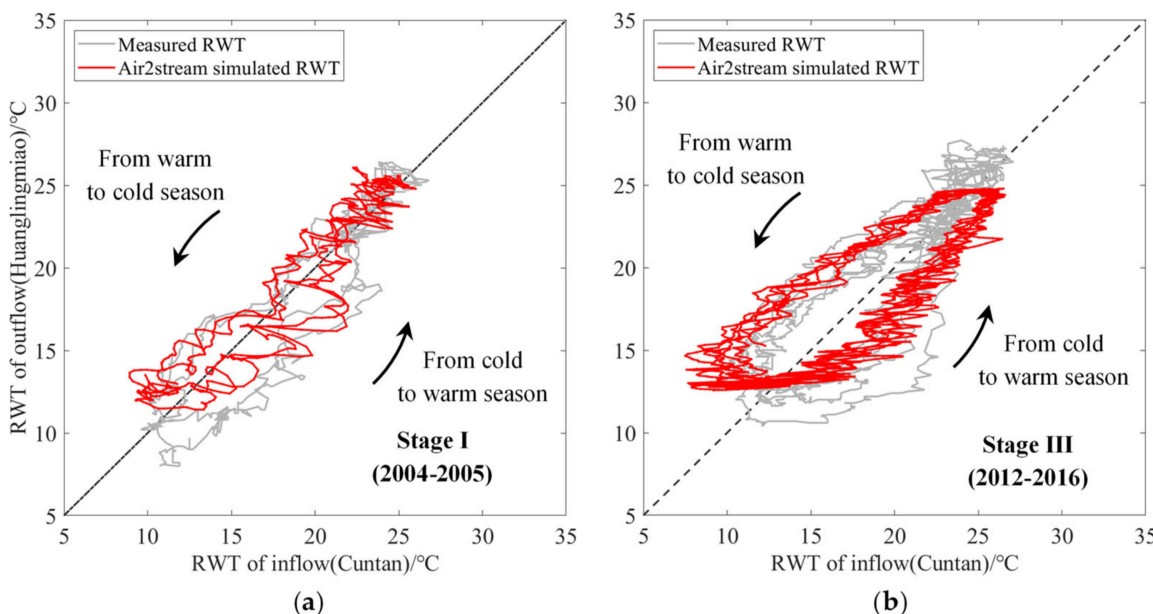

**Figure 8.** Inflow and outflow RWT interrelationship during (**a**) Stage I and (**b**) Stage III.

## 5. Conclusions

In this paper, we proposed a three-stage methodology to retrieve water temperature of rivers or river-type reservoirs by utilising remote sensing TIR data. Taking the Three Gorges Reservoir as a case study, the long-term RWTs in two key stations (Cuntan and Huanglingmiao) were obtained by employing Landsat 5 TM and Landsat 7 ETM+ TIR data. The conclusions are summarised as follows:

- In the proposed method: the key stage of extraction and correction can efficiently decrease the RMSE of RWT by 1 °C, approximately, and the entire method shows a good performance. Therefore, the three-stage methodology is recommended for similar research, especially when retrieving water temperature of rivers or river-type reservoirs using Landsat TIR data.



- Applying the three-stage method, the Landsat-derived daily RWT was suitable and accurate to reveal the changes in RWT under the anthropogenic intervention. It provides a useful tool to obtain the RWT variations in the poorly gauged catchment, which can be used to evaluate and analyse the influence of dams, urban heat island and other human activities over a large spatial and temporal span.
- Two requirements should be met when applying the proposed method and Landsat TIR data to derive RWT, i.e., (1) the river width should be larger than 240 m for Landsat 7 ETM+ (480 m for Landsat 5 TM) and (2) the river water should be well-mixed laterally, and RWT does not change quickly along the river.

To summarise, the three-stage method was a promising approach to estimate the daily RWT and help us understand the thermal regime of rivers or river-type reservoirs under human activities. As Landsat remote sensing images, daily air temperature and discharge were all general data, and apart from the two requirements, there were no more assumptions for the method, the proposed three-stage method could be implemented for similar study areas. (Meanwhile, if you are interested in the code, please contact the corresponding author.)

**Author Contributions:** X.S.: methodology, software, data acquisition, validations, formal analysis, original draft preparation and visualisation. J.S.: conceptualisation, methodology, review and editing, supervision and funding. Z.X. software, resources and review and editing. All authors have read and agreed to the published version of the manuscript.

**Funding:** This research was funded by the National Key R&D Program of China (grant numbers: 2016YFA0600901 and 2016YFE0133700), the National Natural Science Foundation of China (grant number: 51779121) and the State key Laboratory of Hydroscience and Engineering, Tsinghua University (grant number: 2020-KY-03).

**Institutional Review Board Statement:** Not applicable.

**Informed Consent Statement:** Not applicable.

**Data Availability Statement:** Publicly available datasets were analyzed in this study. These data are listed below: (1) Landsat data can be found here: https://earthexplorer.usgs.gov/, (2) air temperature data can be found here: https://data.cma.cn, (3) atmosphere correction parameters can be found here: https://atmcorr.gsfc.nasa.gov.

**Acknowledgments:** Thanks is given for the support from Vera Lu, colleagues, teachers and family members who helped, encouraged and gave advice to the authors during the entire study.

**Conflicts of Interest:** The authors declare no conflict of interest.

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
