# Peer review of "Investigation on River Thermal Regime under Dam Influence by Integrating Remote Sensing and Water Temperature Model"

_water, doi:10.3390/w13020133_

Round 1

Reviewer 1 Report

Review for:

Deriving river thermal regime under the dam

influence by integrating remote sensing and water

temperature model

Summary:

The paper describes a methodology for deriving river water temperatures associated with a Dam on the Yangtze River.  The methodology focuses on the application of Landsat in the three step approach which involves first the calibration of the brightness temperatures, then a regression using in-situ measurements to derive the final SST values. A Radiative Transfer Model is also used to derive the atmospheric corrections. The overall conclusion is that the methodology  is successful in deriving river water temperatures associated with Dams for both inflow and outflow conditions. Overall Root Mean Square differences were around 1 degree Celsius. The paper makes an important contribution. I believe it is publishable after a revision.

Major Changes :

The paper needs a thorough review of the grammar and English. Some examples are listed in the minor changes. Some recommendations:

  • Can the “convolution” methodology be expanded upon? A major component of the three stage approach appears to be the issue land contamination and resolution. I found the discussion is still a bit confusing. Perhaps a table could also help. Even a simple table summaring the three stage approach, inclusive of the different spectral bands on Landsat.
  • My one concern is that after reading the paper one is left with the impression that one really needs to be an expert in Landsat to be able to apply the methodology. Can the following be commented upon (perhaps in the conclusion section). How feasible would it be to apply the methodology to other Dams? The authors should comment upon whether the software is available as open source. If not are there any plans to make it available.

Minor corrections:

Line 59: Please clarify. Landsat did not have that correction? I recommend adding a brief description of the “convolution” method.

Line 65: “but of all them have”. Make plural

Line 225: could you please specify what is mean by “climatologic” problems?

Reference for Marti-Cardona?

Line 270: “while maintaining” not while maintains

Line 310: “established to simulate” not “simulated”.

Line 354: “river length should be” add “be”

Line 409: “separately imported” not “importing”.

Reviewer 2 Report

The writing style of the paper is suitable as a scientific paper. A few revisions are necessary.

  1. Line 194 ; Why is the value 0.9885 is employed in this term?
  2. Line 242 : Boundary effect should be removed. The influence is too large? Please tell us a simple reason why the implementation of boundary effect may cause some troubles?
  3. Figure 7 ; “estimated” and “measured”, the both values are derived rom the satellite image analysis. What is the difference between “estimated “ and “measured”?
  4. What is the most sever deference in the three Stage? Only the water level of the Dam?
  5. Figure 8 ; From the both figures (a) and (b), the influence of the Dam become more representative in Stage III. Is it right?

Reviewer 3 Report

This paper presents a three-stage method to retrieve the daily river or the reservoir water temperature based on the remote sensing techniques and river water modelling. This method was further adopted in the two case study areas including Cuntan and Huanglingmiao. Overall, this paper is well-written and the methods are robust. However, authors should try to improve this paper in two aspects: 

Is it necessary and accurate enough to keep the temperature at three decimal like 1.783C. It is better and meaningful to just the keep the temperature as 1.8C.

In order to enhance the significance of water temperature analysis, authors may also indicate the water cooling performance, with the respect of references including: 

Co-benefits approach: Opportunities for implementing sponge city and urban heat island mitigation. Land use policy86, 147-157.

Changes in the Water Temperature of Rivers Impacted by the Urban Heat Island: Case Study of Suceava City. Water12(5), 1343.

line 102, 'analysing the impact of TGR on the RWT', is this logic?

In Fig.1, some words are not clear.

Conclusion part should be rewritten to generate some implications for future research. Do not need to mention first part or second part.

Round 2

Reviewer 1 Report

The authors have addressed my concerns. I recommend publication.

Reviewer 3 Report

well done. I suggest the acceptance.